# Therapeutic Effect and Safety Evaluation of Naringin on *Klebsiella pneumoniae* in Mice

**DOI:** 10.3390/ijms242115940

**Published:** 2023-11-03

**Authors:** Guanyu Zhao, Qilin Huang, Xiaohan Jing, Lina Huang, Chen Liu, Xiangyi Pan, Zhaorong Li, Sifan Li, Zhengying Qiu, Ruihua Xin

**Affiliations:** 1Lanzhou Institute of Husbandry and Pharmaceutical Sciences of Chinese Academy of Agricultural Sciences (CAAS), Lanzhou 730050, China; zhaogy517@gmail.com (G.Z.); 82101215691@caas.cn (Q.H.); xhanjing306@gmail.com (X.J.); 82101225773@caas.cn (C.L.); 222086002006@lut.edu.cn (X.P.); lizr@st.gsau.edu.cn (Z.L.); lisf@st.gsau.edu.cn (S.L.); 2Engineering and Technology Research Center of Traditional Chinese Veterinary Medicine of Gansu Province, Lanzhou 730050, China; 3Key Laboratory of Veterinary Pharmaceutical Development of Ministry of Agriculture and Rural Affairs of P.R. China, Lanzhou 730050, China; 4State Key Laboratory of Applied Organic Chemistry, School of Pharmacy, Lanzhou University, Lanzhou 730013, China; 220220943870@lzu.edu.cn

**Keywords:** flavonoids, naringin (NAR), *Klebsiella pneumoniae* (Kpn), bacterial pneumonia, anti-inflammatory, drug safety

## Abstract

Critically ill patients with Corona Virus Disease 2019 (COVID-19) often develop secondary bacterial infections that pose a significant threat to patient life safety, making the development of drugs to prevent bacterial infections in the lungs critical to clinical care. Naringin (NAR) is one of the significant natural flavonoids rich in Pummelo Peel (Hua Ju Hong), with anti-inflammatory, antimicrobial, and antioxidant activities, and is commonly used in treating respiratory tract infectious diseases. In this study, the in vitro and in vivo findings revealed that, after *Klebsiella pneumoniae* (Kpn) infection, NAR inhibited overactivation of the nuclear factor kappa-B(NF-κB) signaling pathway in alveolar macrophages of mice, reduced neutrophil (NEs) recruitment, and lowered the induced production of proinflammatory markers, such as Interleukin-6(IL-6) and tumor necrosis factor α(TNF-α). Thus, it suppressed excessive immune responses in the lungs, as well as attenuated the induced pulmonary fibrosis and inflammatory infiltrates. These results suggest that NAR has a preventive effect against Kpn in mice. In addition, the study evaluated NAR’s potential toxicity, demonstrating that NAR is safe at effective doses. These results suggested that NAR effectively reduces excessive inflammatory damage in the lungs induced by Kpn and enhances the body’s ability to clear bacteria. Therefore, NAR may be an effective and safe healthcare drug for preventing and caring for bacterial pneumonia.

## 1. Introduction

In recent years, COVID-19 has ravaged the world, placing a heavy burden on the global healthcare system [1,2]. After viral infection, host cells trigger inflammatory responses and release excessive inflammatory mediators, leading to dysregulation of the body’s immune response and creating opportunities for secondary bacterial infections [3]. Bacterial infections associated with COVID-19, such as pneumonia, sepsis, and meningitis, may be the leading cause of death in patients [4]. A total of 43 dominant-negative strains of pathogenic bacteria were isolated from 65 patients with COVID-19, mainly *Klebsiella pneumoniae* (Kpn), coagulase-negative bacteria, *Pseudomonas aeruginosa*, and *Acinetobacter baumannii* [5,6]. With the decreasing virulence and lethality of COVID-19, the focus of attention has gradually favored the rehabilitation of secondary pneumonia and the regulation of the body’s immune function.

Natural medicines have immunomodulatory effects on the body and are of broad interest, especially in treating inflammatory diseases. Pummelo Peel (Hua Ju Hong) is mainly derived from *Exocarpium Citri Grandis*, the dried unripe fruit peel of Citrus grandis ‘Tomentosa’, or *Citrus grandis* (L.), which functions in dissolving phlegm, removing dampness to benefit the stomach, and eliminating food. As a medicinal and food homologous traditional Chinese medicine, Pummelo Peel has been increasingly used in medicine and healthcare food. However, due to its low production and limited production areas, the general application of Pummelo Peel has been limited to a certain extent. However, flavonoids are the practical parts of Pummelo Peel, among which the content of dihydroflavonoid naringin (NAR) can reach up to more than 98% [7], which is the main active ingredient in Pummelo Peel [8]. In addition, besides Pummelo Peel, NAR is also widely found in the rinds of grapefruit, tangerine, orange, and other plants of the Rutaceae family, making NAR not only have traditional pharmacological traceability but also an advantage of abundant resources.

Studies have shown that NAR has multiple pharmacological effects, such as anti-inflammatory, antibacterial, antioxidant, antitumor, and antiosteoporosis [4,9,10]. Especially in treating inflammatory lung diseases, NAR can exert a medicinal value comparable to Pummelo Peel [11]. Studies have shown that NAR plays a potential anti-inflammatory role in carrageenan-induced pleurisy and inhibits the expression of inflammatory cytokines in rats with chronic obstructive pulmonary disease (COPD) induced by cigarette exposure [12,13]. NAR has also been shown to suppress eosinophilic airway inflammation in ovalbumin-induced asthma and attenuate acute neutrophil (NEs) infiltration induced by airway hyperreactivity. These reports confirm the therapeutic role of NAR in lung disease, but the therapeutic effect on pneumonia caused by Kpn infection remains unclear.

In this study, we used Kpn to induce a mouse pneumonia model and a mouse alveolar macrophage inflammation model. We investigated the protective effect of NAR intervention against Kpn-induced pneumonia. Moreover, the safety of NAR was evaluated. This study provided theoretical support for NAR as a treatment for bacterial pneumonia and also provided a research basis for the development of NAR as a healthcare product.

## 2. Results

### 2.1. Therapeutic Effect of NAR on Mice with Kpn

The specific drug administration and challenge schemes are shown in Figure 1A. The result indicates that the model group showed a decreasing trend in body weight after Kpn infection. Still, NAR treatment inhibited weight loss, with the most apparent inhibitory effect in the high-dose group (Figure 1B). After modeling, food consumption in the model group and the NAR treatment groups decreased significantly and then gradually recovered. Recording of the daily food intake revealed that it was lowest in the model group and significantly higher in the NAR groups than in the model group (Figure 1C). Since the number of bacteria in the lungs is positively correlated with the severity of pneumonia, so we investigated the number of Kpn colonies in the bronchoalveolar lavage fluid (BALF) in our mice. The results of streaking on agar plates showed that colony growth was most dense in the model group and was comparatively less dense in the NAR-treated groups, with almost no colony growth at the high dose of 80 mg/kg body weight, indicating that the inhibitory effect was most pronounced at this concentration (Figure 1F). The result of the bacterial-dilution coating-plate method showed that no Kpn colonies grew from the BALF of mice in the control group, indicating that Kpn did not infect them. The number of Kpn colonies was highest in the BALF from the model group (5.87 × 10^7^ colony-forming units (CFU)) and dropped significantly to 3.86, 2.07, and 0.27 × 10^7^ CFU in response to NAR in the low-, medium- and high-dose groups, respectively (Figure 1D). Furthermore, we examined the level of C-reactive protein (CRP) in the BALF to further assess the anti-infective efficacy of NAR in vivo. As shown in Figure 1E, CRP was significantly increased in the BALF of mice in the model group compared with those in the control group (*p* < 0.05) but was significantly decreased after NAR intervention in a dose-dependent manner, with a particularly pronounced reduction in the high-dose group, which was not significantly different from that in the control group (*p* > 0.05).

### 2.2. Effect of NAR on the Pathological Injury of the Lung in Mice

After the mice were executed, postmortem examinations were performed. The control mice were found to have healthy pink lungs. In contrast, the model mice had blunt edema with the formation of red masses and spots that gradually developed into scattered inflammatory lesions in both lungs. However, the NAR administration group showed only localized areas of mild solid change and a few sites of blood spots, and visual observation indicated the lesion was in remission, with the most pronounced relief effect in the high-dose group. H&E staining and Masson staining were used to determine the effects of NAR on the pathological blood vessels and bronchi alterations in Kpn-induced mice. As shown in Figure 2, in the Model group, red blood cells and flocculent secretions exuded into the bronchioles, and there was apparent inflammatory cell infiltration around the bronchus. Loose edema and fibrosis formed around blood vessels and bronchi, and compared with the model group; the NAR treatment groups exhibited varying degrees of alleviated lung tissue injury, most prominently in the high-dose group, in which the tracheal structure was complete. The tracheal mucosal epithelial cells were more neatly arranged, and there was the disappearance of the inflammatory exudate filling the bronchioles, decreased infiltration of surrounding inflammatory cells, and a decrease in the degree of fibrosis.

### 2.3. Regulation Effect of NAR on Pulmonary Inflammatory Infiltration in Mice

Myeloperoxidase (MPO) is mainly peroxidase released by neutrophils (NEs), and the expression level of MPO can reflect the degree of inflammatory cell infiltration and indirectly reflect the level of local inflammation. As shown in Figure 3A, we found a significant increase in MPO content in the lung tissues of the model group compared with that in the control group, and a corresponding decrease in MPO content in the NAR treatment groups. Immunohistochemistry was used to characterize the expression of MPO in lung tissues for exploring the protective effect of NAR on lung inflammatory injury after Kpn infection. MPO-immunoreactive cells (MICs) were mainly distributed in the bronchial epithelium and the alveolar septum. There were small numbers of MICs in the control group, and the comparatively high numbers of MICs in the bronchial wall and septum of the lung in the model group showed a decreasing trend following different doses of NAR intervention (Figure 3D). These results were confirmed by ImageJ software 1.52a analysis of the integrated optical density (IOD) (Figure 3B) and counting of MICs in lung tissue (Figure 3C). Thus, NAR significantly alleviated the Kpn-induced infiltration of large numbers of NEs in lung tissue, thereby exerting an anti-inflammatory effect.

### 2.4. Regulation of NAR on the Classical Inflammatory Signaling Pathway

Macrophages are critical immune cells that elicit the body’s immune response, as they are capable of capturing, processing, and presenting antigens. The NF-κB signaling pathway is the main pathway for the regulation of inflammatory factors and stimulation of macrophage activation. This cascade of reactions is successively phosphorylated by IkBα, and inhibitor of kappa B kinase (IKK). An essential protein of NF-κB, P65, ultimately triggers the transcription of many proinflammatory factors, including IL-1β, IL-6, and TNF-α, thus promoting the recruitment and activation of lymphocytes and further amplifying the inflammatory response. The progressive phosphorylation of IKK, IκBα, and P65 is a key step in the activation of the NF-κB signaling pathway, which can reflect the activation level of NF-κB. Therefore, we analyzed the phosphorylation levels of these critical proteins in mouse lung tissue by Western blotting and found that Kpn infection significantly increased the ratios of phosphorylated (p)-IκBα/IκBα, p-IKKα-β/IKKα, p-IKKα-β/IKKβ, and p-P65/P65. Compared with the model group, the ratios of p-IκBα/IκBα, p-Ikkα-β/IKKα, p-Ikkα-β/IKKβ, and p-P65/P65 were significantly decreased in the medium- and high-dose NAR groups (*p* < 0.05) (Figure 4A,B). These results indicated that NAR could reduce the phosphorylation levels of IkBα, IKK, and NF-κB P65 protein. The expression of p-P65 in mouse lung tissue was also detected by immunohistochemistry. In the model group, there were a large number of nuclear hyperchromatic cells in the bronchial wall and pulmonary septum, indicating that a large amount of NF-κB P65 entered the nucleus under induction by Kpn; the numbers of these cells were decreased to varying degrees in the NAR treatment groups (Figure 4C), indicating that NAR administration reduced the nuclear translocation of p-P65. These results were confirmed by software analysis of the IOD (Figure 4D) and counting p-P65 immunoreactive cells (Figure 4E) in lung tissue.

### 2.5. Medication Safety Evaluation of NAR

To further ensure the safety of NAR administration, we gave NAR to mice at high doses and evaluated its potential toxicity. The body weights of three groups (control group, 80 mg/kg, and 160 mg/kg NAR dose groups) of mice were recorded during 7 d of administration. As shown in Figure 5A, continuous administration of NAR at high doses did not cause significant changes in the body weights of control group mice (*p* > 0.05). Upon completion of the experiment, the organs and blood of mice in each group were collected and tested. The blood biochemical results showed no significant differences (*p* > 0.05) in the serum renal function indexes GREA, UREA, and hepatic function indexes TG, TC, ALT, AST, TBil, ALP, TP, and ALB of mice in the NAR group as compared with those in the control group (Table 1). These conclusions tentatively exclude the potential toxicity of high-dose NAR to the liver and kidney.

Apart from that, the different visceralities of the mice were weighed to calculate the corresponding organ indices, as shown in Figure 5B–E. The viscera indices of the heart, liver, spleen, and kidney of mice in all groups were not significantly different from those of the control group (*p* > 0.05). Meanwhile, H&E staining showed that in both NAR dose groups, the cellular structure was intact in the heart, liver, spleen, lungs, and kidneys, and no significant lesions were observed compared with the blank group. In the heart tissue, the myocardial fibers were well arranged and continuous, and the nuclei were primarily located in the center of the myofibers without edema or vacuoles. In the liver tissue, the hepatic lobules were structured, and the hepatocytes were arranged in a neat radial pattern along the central vein. In the splenic tissue, the boundaries of red and white pulp were clear, the trabecularity of the spleen was intact, and the cellular morphology was normal. In the lung tissue, the alveolus and alveolar sacs were structurally intact and well formed, the cells of the bronchial wall were tightly arranged, and no inflammatory secretion or hemorrhage was seen in the bronchial lumen or alveolar lumen. The renal cortex and medulla were seen in the kidney tissue, and the glomeruli and renal capsule were intact structurally without significant congestion (Figure 5F).

### 2.6. Effect of NAR on the Viability and Production of Proinflammatory Cytokines in the MH-S Cell Line

The potential toxicity of NAR was investigated by morphological observation of cells and cytotoxicity assays. After a 24 h coincubation of NAR and MH-S cells, no swelling, breakage, shrinkage, or extravasation of the cell contents was observed under a light microscope (Figure 6A). The latent cytotoxicity of NAR at six different concentrations (5, 10, 15, 20, 25, or 30 μg/mL) on MH-S was evaluated by the 3-(4,5-dimethylthiazol-2-yl)-2,5-diphenyltetrazolium bromide (MTT) assay. There was no noticeable change in cell viability within 24 h of NAR treatment, indicating that NAR did not cause latent cytotoxicity in MH-S (Figure 6B). The effect of each concentration of NAR on IL-1β, IL-6, and TNF-α secretion in uninfected MH-S cells was determined by ELISA kits (Solarbio Science and Technology Co., Ltd., Beijing, China). NAR did not affect the secretion of IL-1β, IL-6, and TNF-α in uninfected MH-S cells (Figure 6C–E). The results indicate that NAR does not produce irritation to MH-S.

### 2.7. Effect of NAR on the Expression of Proinflammatory Cytokines in Inflammatory Cell Models

To determine the effect of NAR on proinflammatory cytokine production, we established an inflammatory cell culture model of MH-S cells infected with Kpn. ELISA of culture supernatants revealed that Kpn infection induced high levels of IL-6 and TNF-α secretion in MH-S cells at 2 h post infection: 17.09 and 14.40 pg/mL, respectively (*p* < 0.05). Pretreatment of MH-S cells with different doses of NAR (10, 20, and 30 μg/mL) significantly reduced the secretory overproduction of IL-6 and TNF-α under Kpn infection compared with untreated cells (Figure 7A,B). In addition to proinflammatory factors, lactate dehydrogenase (LDH) is an essential criterion for cellular inflammation models. Compared with the control group, Kpn-induced LDH levels were significantly increased in MH-S cells and decreased after NAR treatment, indicating that NAR has a reversal effect on Kpn-induced cellular inflammatory damage (Figure 7C). The expression levels of these proinflammatory factors in MH-S cells were verified at the genetic level. Consistent with these results, Kpn infection significantly increased the mRNA levels of IL-6 and TNF-α in MH-S cells (*p* < 0.05) (Figure 7D,E). Treatment with NAR at 10, 20, or 30 μg/mL was able to decrease the mRNA expression of IL-6 in Kpn-infected MH-S cells, while only the 30 μg/mL dose had a significant effect on the production of TNF-α. IL-10 reportedly exerts anti-inflammatory effects by antagonizing the proinflammatory effects of other cytokines. In the present study, all NAR dose groups promoted the secretion of IL-10 in MH-S cells (Figure 7F), suggesting that NAR achieves its anti-inflammatory effects by inhibiting the secretion of proinflammatory cytokines and increasing the secretion levels of anti-inflammatory factors in MH-S.

### 2.8. Regulation of the NF-κB Signaling Pathway by NAR

Inflammatory factors are closely related to the overactivation of NF-κB. Since the degree of phosphorylation of essential proteins in the NF-κB signaling pathway directly reflects the extent to which the pathway is activated, we examined the effect of NAR at different doses on the phosphorylation of these proteins in MH-S cells using Western blotting. Similar to the results of the in vivo assay, Kpn infection of MH-S cells promoted the expression of the supracellular transmembrane protein toll-like receptor 4(TLR4), which then induced the phosphorylation of IKKα, IKKβ, and IκBα in the IKK complex, resulting in a significant increase in the phosphorylation level of NF-κB P65 at 2 h post Kpn infection (*p* < 0.05). The data indicate that entry of P65 into the nucleus was promoted, ultimately allowing a large number of proinflammatory factors to be released and leading to cellular inflammatory damage. The phosphorylation levels of IKKα, IKKβ, IκBα, and P65 were decreased after NAR pretreatment at 20 or 30 μg/mL compared with those in the model group (Figure 8A). The levels of protein expression were calculated using ImageJ software 1.52a, which revealed that the ratios of p-IκBα/IκBα, p-IKKα-β/IKKα, p-IKKα-β/IKKβ, and p-P65/P65 were significantly decreased after NAR treatment (Figure 8B–F). These data suggested that NAR can negatively regulate the phosphorylation levels of vital nodal proteins in the NF-κB signaling pathway to exert anti-inflammatory effects.

## 3. Discussion

Respiratory problems, such as lung infections caused by the outbreak of COVID-19 pneumonia, pose a severe test to the healthcare system. It has been found that patients infected with COVID-19, especially those with severe disease, are more prone to secondary bacterial infections, with the percentage of bacterial secondary infections caused by Kpn approaching 10% [13]. Currently, antibiotics are still the preferred antimicrobial drugs in the clinic. However, patients with Kpn are weaker and more prone to side effects. Therefore, the need for drugs with anti-inflammatory and immunomodulatory effects that can be used in daily healthcare is even more significant.

Flavonoid derivatives are one of the most researched and used natural active ingredients with various biological activities such as anti-inflammatory, anticancer, antiviral, and antioxidant. Recent studies have found that flavonoids not only elevate the body’s resistance to COVID-19, but also participate in the clearance of residual viruses from the body after illness, which has excellent potential for treating and rehabilitating COVID-19 in healthcare [14,15]. Flavonoids of Brassicaceae origin, such as NAR and naringenin, can better alleviate the morbidities, such as pulmonary fibrosis, that occur during virus infection [16,17,18]. Moreover, NAR was found to have potent anti-inflammatory and antibacterial bioactivities, especially for cough, sputum, and lung infections [19,20,21,22]. Our study found that NAR inhibited macrophage activation and reduced lung aggregation of NEs; decreased secretion of proinflammatory factors and reduced lung invasion of Kpn; ameliorated lung congestion and pulmonary bronchial fibrosis; and reversed the decrease in body weight and food intake in mice. Its mechanism of action may be phosphorylation inhibition of P65 proteins, IKK, and IκB, along with downregulation of the NF-κB signaling pathway, thereby reducing the pathway’s excessive activation.

Kpn is the most common causative agent of pneumonia, and under pathological conditions, Kpn invades and adheres to the lungs, releasing a variety of virulent molecules, mainly LPS. These external stimuli first cause the aggregation and activation of macrophages, and the activated macrophages initially process antigenic molecules and recruit immune cells such as NEs, leading to an increase in the secretion of MPO [23]. Additionally, significant secretion of proinflammatory factors such as IL-1β, IL-6, and TNF-α leads to an immune imbalance, which causes an inflammatory response in the lungs, leading to the development of pneumonic injury [24].

TLR4, a natural immune receptor expressed on the surface of AMs, effectively recognizes PAMPs and is the central receptor for LPS. LPS binding to TLR4 activates NF-κB to promote the expression of inflammatory cytokines [25]. Studies have shown that macrophage polarization is vital in regulating the inflammatory response. Recently, many natural medicines have been shown to inhibit M1 macrophage polarization by inhibiting the TLR4/NF-κB signaling pathway. For example, berberine competitively inhibits the binding of TLR4 and MyD88 to inhibit the TLR4/NF-κB signaling pathway, thus inhibiting the polarization of M1 macrophages [26]. Similarly, quercetin downregulates the expression of NF-κB and IRF5, thereby inhibiting the activity of upstream TLR4/MyD88 and subsequent M1 polarization [27]. These findings strongly support the critical role of the TLR4/NF-κB signaling pathway in the polarization of M1 macrophages. In this study, NAR reduced the phosphorylation levels of three essential proteins (IKKα/β, IκBα, and P65) in the TLR4/NF-κB signaling pathway in an animal Kpn pneumonia model and had a specific inhibitory effect on the signaling cascade in M1 macrophages. NAR was also found to reduce Kpn-induced inflammatory cytokine release, inhibit phosphorylation of critical proteins in the TLR4/NF-κB signaling pathway, and prevent overactivation in vitro. After phosphorylation, the P65 protein enters the nucleus. It binds to the specific DNA target site B [28,29] to trigger the expression of downstream target genes, thus regulating the body’s immune and inflammatory responses. Therefore, the degree of nuclear translocation of the p-P65 protein reflects the degree of activation in the NF-κB signaling pathway to a certain extent [30]. Our results suggest that NAR can significantly inhibit the translocation of NF-κB P65 from the cytoplasm to the nucleus after Kpn induction, thereby exerting an anti-inflammatory effect.

Flavonoids are generally considered safe, but toxic responses such as hepatotoxicity and nephrotoxicity can be observed in long-term use. Therefore, the safety of NAR was examined in this experiment. Our results showed that at 160 mg/kg (twice the high dose), the indicators of TP, ALB, GREA, and UREA in the blood of mice were in the normal range, and no obvious abnormality was observed in the pathological tissues of the heart, liver, spleen, lungs, or kidneys. The results indicated that NAR has no potential toxic effects on the organism at both in vivo and in vitro levels under the effective dose, which guarantees the further use of NAR as a safe drug in the clinic.

## 4. Materials and Methods

### 4.1. Chemicals and Reagents

NAR was purchased from Sigma Aldrich Chemical (St. Louis, MO, USA. HPLC ≥ 98%, chemical formula C_27_H_32_O_14_, molecular weight 580.54, CAS No. 10236-47-2), Dulbecco’s modified Eagle’s medium (DMEM) and trypsin were purchased from JS Biosciences Co., Ltd. (Lanzhou, Gansu Province, China). Fetal bovine serum (FBS) and penicillin–streptomycin antibiotics were purchased from Thermo Fisher Scientific (Waltham, MA, USA). The LDH assay kit, ELISA kits for IL-6, TNF-α, and nucleoprotein extraction were purchased from Solarbio Science and Technology Co., Ltd. (Beijing, China). ELISA kits for IL-1β and MPO were purchased from Jianglai Industrial Limited by Share Ltd. (Shanghai, China). Cell Counting Kit-8 (CCK-8) was purchased from Labgic Technology Co., Ltd. (Beijing, China). The RNA extraction kit, TRIzol^®^ Reagent RT-PCR kit, and SYBR^®^ green PCR master mix were bought from TaKaRa (Tokyo, Japan). The primary antibodies against IKKα, IKKβ, p-IKKα/β, NF-κB P65, p-P65, IκBα, p-IκBα, GAPDH, Lamin B, HRP-conjugated goat antirabbit IgG, HRP-conjugated goat antimouse IgG and Alexa Fluor 488 labeled antimouse IgG were obtained from Cell Signaling Technology, Inc. (Beverly, MA, USA). Bovine serum albumin (BSA) and the ECL detection kit were supplied by Thermo Fisher Scientific (MA, USA). McConkey’s medium was purchased from Sigma-Aldrich (Merk, Germany). All the other reagents employed in the study were of analytic grade and are commercially available unless otherwise stated.

### 4.2. Establishment of Animal Model and Treatment of NAR

Sixty Kunming mice (18–22 g) were provided by the Lanzhou Veterinary Institute of CAAS (License No. SCXK Gansu 2022-0009). Animal welfare statement: institutional ethical and animal care guidelines were observed, and all experimental processes were conducted according to the China Guide for the Care and Use of Laboratory Animals (protocol number: IACUC-157031). A total of 5 groups were divided into a control group (gavage with water), model group (gavage with water + 10^8^ CFU/mL Kpn infection), NAR low-dose group (gavage with 20 mg/kg NAR + 10^8^ CFU/mL Kpn infection), NAR medium-dose group (gavage with 40 mg/kg + 10^8^ CFU/mL Kpn infection), and NAR high-dose group (gavage with 80 mg/kg + 10^8^ CFU/mL Kpn infection) randomly (n = 12). Before infection of Kpn, mice from drug treatment groups received NAR of the corresponding concentrations by gavage once a day for eight consecutive days, and the control group and model group were given normal saline. On day 7, the mice were narcotized by the intraperitoneal injection of sodium pentobarbital (50 mg/kg b.w.). The Kpn was injected into the lungs of the mice through tracheal intubation (100 μL for each body; the concentration of a bacterial solution was 1 × 10^8^ CFU/mL); the control group was injected with an equal volume of sterile normal saline. Three days after infection, blood was collected from the orbital venous plexus for physiological analysis, and then the mice were anesthetized and sacrificed; different samples were taken for the following steps.

### 4.3. In Vivo Safety Evaluation of NAR

Thirty Kunming mice (18–22 g) were divided (n = 10) randomly into a control group, NAR group (80 mg/kg), and NAR group (160 mg/kg). After oral administration of the drug for seven consecutive days, the mice were executed, and samples of heart, liver, spleen, lungs, kidneys, and blood were collected for subsequent experiments.

### 4.4. Bacteria, Cellular Inflammation Models, and NAR Treatment

Klebsiella Pneumoniae (Kpn) kept in our laboratory was obtained from a community hospital and verified by sequencing; LB medium (1% tryptone, 0.5% yeast exact, 1% NaCl) was used for culture at 37 °C before use. Mouse alveolar macrophage (MH-S) cells were acquired from the Procell Life Science and Technology Co., Ltd. (Wuhan, China). and cultured in DMEM, including 1% penicillin–streptomycin as well as 10% FBS in the humidified atmosphere with 5% CO_2_ at 37 °C. After cell confluency reached about 70–80%, cells were trypsinized and subcultured. The cells were pretreated with NAR (10, 20, and 30 µg/mL) for 12 h and then stimulated with Kpn bacterial solution (1 × 10^8^ CFU/mL) for 1 h. After removing the bacterial solution, the culture was continued as usual for 12 h. The cells were collected for Western blotting and RT-PCR, and the supernatant was collected for ELISA.

### 4.5. Blood Analysis

For blood physiological analysis, the whole blood was analyzed using an Abaxis Blood Chemistry Analyzer (Abaxis, Inc., Union City, CA, USA) according to the manufacturer’s instructions.

### 4.6. Histopathologic Examination

All tissues of mice treated with different NAR doses were harvested, fixed in 4% formaldehyde for 24 h, embedded in paraffin, and then sliced into 4 μm thick sections with a slicer. The sections were stained with hematoxylin and eosin (H&E), and lung tissues were stained with Masson. A microscope (DM 4000B, Leica, Germany) was used for histological observation.

### 4.7. Immunohistochemistry (IHC) Assay

The paraffin sections were routinely dewaxed and thoroughly cleaned and then placed in 0.01 mol/L sodium citrate buffer solution to repair the antigen by microwave for 25 min. After washing with PSB, the paraffin sections were sealed with fetal bovine serum for 1 h, 1:100 diluted primary antibody (MPO and P65) was added, and then incubated at 4 °C overnight. After the PSB was washed, the HRP-labeled secondary antibody was dripped and incubated at room temperature for 40 min and then placed in the newly configured DAB solution for color rendering. The microscope (DM 4000B, Leica, Germany) was used for image acquisition, and the integrated optical density (IOD) was calculated by ImageJ software 1.52a.

### 4.8. Quantitative Real-Time PCR Analysis of the Expression of Inflammatory Factors

Total RNA was isolated using TRIzol (15596026, Invitrogen Life Technologies, Carlsbad, CA, USA). mRNA was synthesized using an RT reagent kit with a gDNA Eraser (RR047Q, Takara, Dalian, China). qPCR analysis was performed using TB Green^®^ Premix EX Taq™ II (R075A, Takara, Dalian, China) on ABI Q5 (Thermo Fisher Scientific, USA). Relative mRNA abundance was calculated using the 2^−ΔΔCt^ method. The gene-specific oligonucleotide primers used for qPCR *β-actin* F: GGTCACCAGGGCTGCTTT, *β-actin* R: ACTGTGCCGTTGACCTTGC; *IL-6* F: GCAGGCAGTATCACTCATTGT, *IL-6* R: GCATTGGAAATTGGGCTAGG; *TNF-α* F: AAGGGAGAGTGGTCAGGTTGC, *TNF-α* R: CAGAGGTTCAGTGATGTAGCG; *IL-10* F: GCTCCTAGAGCTGCGGACTGC, *IL-10* R: TGCTTCTCTGCCTGGGGCATCA.

### 4.9. ELISA Analysis of the Expression of Inflammatory Factors

Collecting samples in vivo and in vitro (cell supernatant, lung tissue, and BALF), a commercially available LDH assay kit (Solarbio Science and Technology Co., Ltd., Beijing, China) for detecting the degree of pyroptosis, while C-reactive protein (CRP), IL-1β, IL-6, and TNF-α reflected the level of inflammation. All procedures were performed according to the manufacturer’s instructions.

### 4.10. Western Blotting Analysis

The collected lung tissue homogenate or cells were lysed with buffer A PMSF and then centrifuged at 15,000× *g* for 10 min at 4 °C; the supernatants were harvested as the protein extracts, and the concentration of the protein specimens was measured according to bicinchoninic acid assay (BCA). After the preparation of the protein extracts, identical quantities of the protein specimens were detached in 8–12% SDS-PAGE and then electrotransferred into the polyvinylidene difluoride membrane (Millipore, Billerica, MA, USA). The membranes were sealed with 5% skimmed milk at room temperature for 1.5 h and incubated at 4 °C overnight by the following specific primary antibodies against IKKα (1:1000), IKKβ (1:1000), p-IKKα/β (1:1000), NF-κB P65 (1:1000), NF-κB p-P65 (1:1000), IκBα (1:1000), and p-IκBα (1:1000). Then, HRP-conjugated goat antimouse IgG secondary antibody (1:5000) or HRP-conjugated goat antirabbit IgG secondary antibody (1:5000) was appended and incubated at 37 °C for 1 h. Ultimately, the membranes were visualized by the chemiluminescence detection kit and examined with a Biosciences Imager. GAPDH or Lamin-B served as the internal control normalized against the overall proteins. Further, the phosphorylated proteins were normalized against their respective total proteins. The densities of the bands were quantified by ImageJ software 1.52a and calculated by the normalization against the densitometric value of the internal loading control.

### 4.11. Statistical Analysis

GraphPad Prism 8.0 (GraphPad Software Inc., San Diego, CA, USA) was used for graphing, and SPSS 24.0 was used for statistical analysis (SPSS Inc., Chicago, IL, USA). Values were expressed as mean ± SEM. Differences between groups in body-related parameters were analyzed using a one-way analysis of variance (ANOVA) and Tukey’s post hoc test. For all tests, a probability value (*p*) of less than 0.05 (*p* < 0.05) was considered statistically significant. Different lowercase letters on the error bars indicate statistically significant differences (*p* < 0.05).

## 5. Conclusions

In conclusion, the present study demonstrated that NAR enhanced the resistance to Kpn and attenuated the symptoms of Kpn-induced pneumonia in mice. Specifically, NAR reduced the phosphorylation of essential proteins in the macrophage NF-κB signaling pathway, decreased the aggregation of NEs in the lungs, lowered the expression of IL-6 and TNF-α, and attenuated lung tissue damage induced by the excessive inflammatory cytokines. Moreover, using NAR in effective doses is safe for the organism. In conclusion, NAR is a safe and effective drug to alleviate bacterial infections in the lungs and can be a plant-based health product that alleviates pneumonia. 

## Figures and Tables

**Figure 1 ijms-24-15940-f001:**
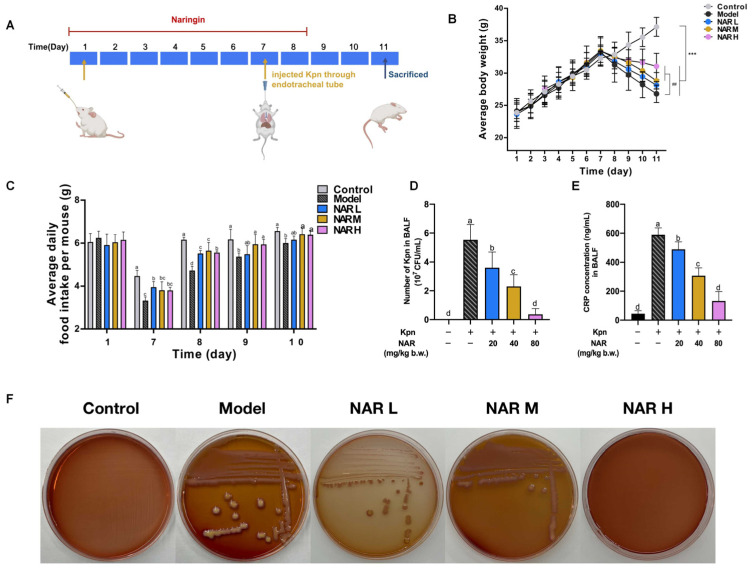
The therapeutic effect of NAR on Kpn-induced pneumonia in mice. (**A**) Specific drug administration and challenging schemes. The NAR administration groups (20, 40, or 80 mg/kg·bw) were able to reverse the reduction in body weight (**B**) and food intake (**C**) in mice due to Kpn pneumonia. (**D**) Bacterial-dilution coating-plate results showed that NAR reduces the number of Kpn colonies in the BALF of mice. (**E**) C-reactive protein (CRP) amount in BALF. (**F**) Results of streaking on agar plates showed that NAR reduces the number of Kpn colonies in the alveolar lavage fluid (BALF) of mice. (Different lowercase letters indicate statistically significant differences between groups *p* < 0.05, while the same lowercase letters indicate no significant differences between groups *p* > 0.05, the same below. Only in (**B**), *** < 0.001 vs. control group; ## < 0.01 vs. model group).

**Figure 2 ijms-24-15940-f002:**
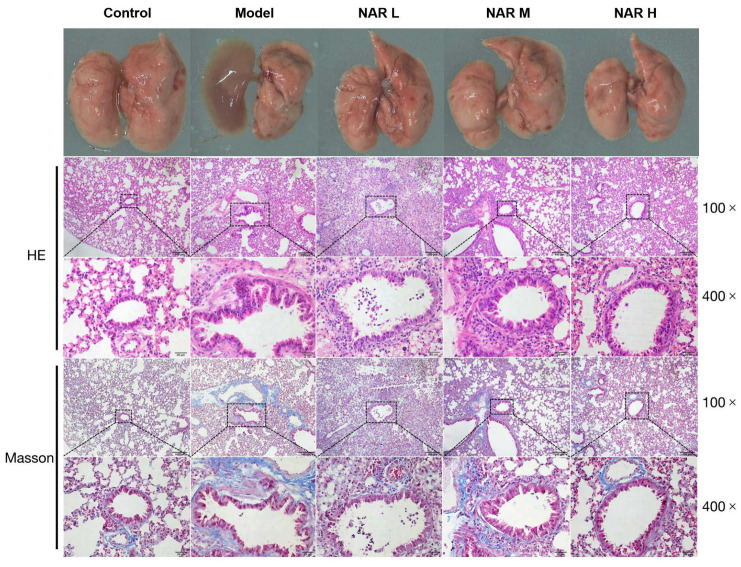
Effect of NAR on Kpn-induced damage in mice lung. Physiological autopsy results showed that NAR alleviated surface inflammatory injury in the lungs of model mice, and H&E and Masson staining demonstrated that NAR reduced Kpn-induced inflammatory cell infiltration and lung fibrosis. Scale bar: 100 μm (100×), 20 μm (400×).

**Figure 3 ijms-24-15940-f003:**
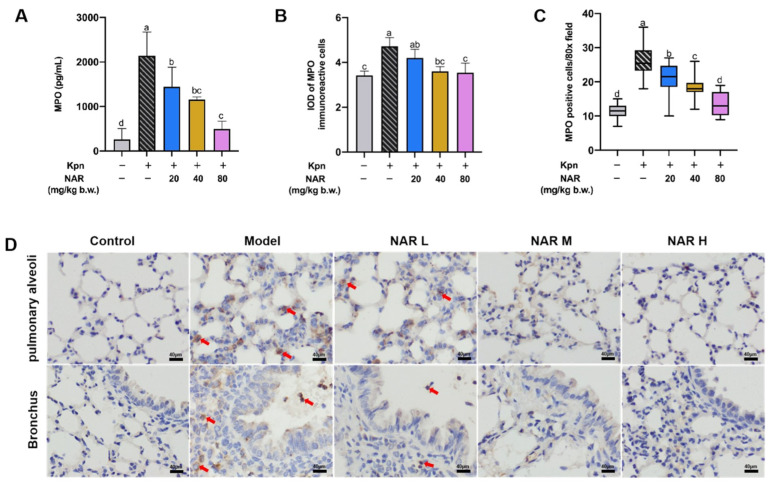
Effect of NAR on Kpn-induced inflammatory infiltration in the lung of mice. (**A**) NAR (20, 40, or 80 mg/kg·bw) reduces the elevation of MPO in lung tissue caused by Kpn pneumonia. (**D**) Immunohistochemical examination of MPO in lung tissue, the red arrows show MPO-immunoreactive cells. (**B**,**C**) Integrated optical density (IOD) of MPO-immunoreactive cells and positive cell count. Scale bar: 40 μm. (The red arrows point to the MPO-immunoreactive cells (MICs).

**Figure 4 ijms-24-15940-f004:**
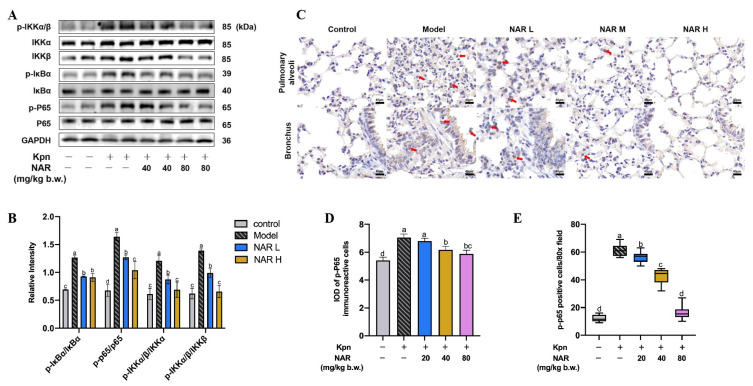
Effect of NAR on NF-κB signaling pathway in the lung tissue. (**A**) Western blotting analysis of the NF-κB signaling pathway. (**B**) Relative quantitative protein expression levels indicated that NAR intervention reduced the protein expression of p-IκBα/IκBα, p-IKKα-β/IKKα, p-IKKα-β/IKKβ, and p-P65/P65 in lung tissue. (**C**) In the immunohistochemical examination of NF-κB P65 in lung tissue, the red arrows show p-P65 immunoreactive cells. (**D**,**E**) Integrated optical density (IOD) of p-P65-immunoreactive cells and positive cell count. Scale bar: 40 μm.(The red arrows point to the p-P65 immunoreactive cells).

**Figure 5 ijms-24-15940-f005:**
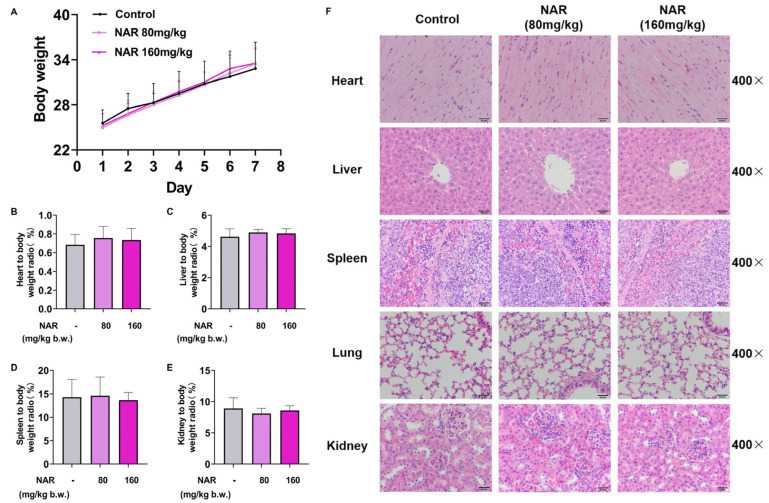
Safety evaluation of NAR in vivo. NAR-administered mice did not exhibit significant changes in body weight (**A**) and visceral indices of heart (**B**), liver (**C**), spleen (**D**), and kidney (**E**) at the high doses (80 mg/kg, 160 mg/kg b.w.). (**F**) H&E staining showed no apparent pathologic damage in the heart, liver, spleen, lungs, or kidneys in the NAR group.

**Figure 6 ijms-24-15940-f006:**
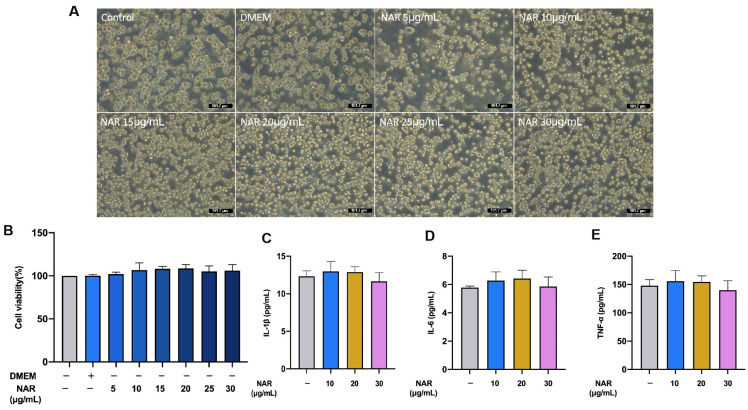
The effect of NAR on the viability and production of proinflammatory cytokines and chemokines in MH-S. In each experimental group, MH-S cells were treated with NAR (5, 10, 15, 20, 25, or 30 μg/mL) or PBS (10 μL). No swelling, breakage, shrinkage, or extravasation of the cell contents was observed by light microscope (**A**). The cell viability was determined by the MTT assay (**B**). MH-S cells were treated with different concentrations of NAR (5, 10, 15, 20, 25, or 30 μg/mL) for 24 h. The concentrations of IL-1β, IL-6, and TNF-α in the supernatant of MH-S cultures were analyzed using ELISA (**C**–**E**). Results were analyzed by one-way ANOVA with Tukey’s post hoc test. No significant difference was observed compared to the control group (*p* > 0.05). The scale bar is 101.7 μm.

**Figure 7 ijms-24-15940-f007:**
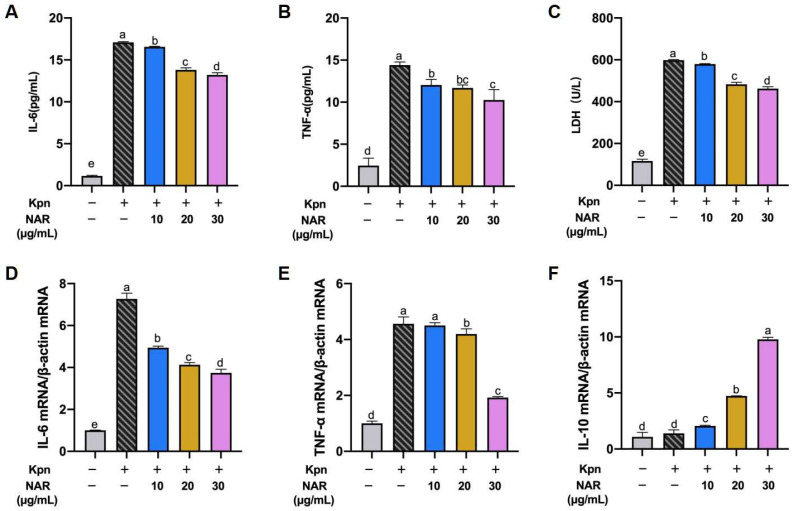
Effect of NAR on Kpn-induced proinflammatory cytokines in MH-S. NAR (10, 20, and 30 μg/mL) significantly reduced the secretory overproduction of IL-6 (**A**), TNF-α (**B**), and LDH (**C**) due to Kpn infection compared to the model group, based on the ELISA method. NAR (10, 20, and 30 μg/mL) significantly reduced the expression quantity of IL-6 (**D**) and TNF-α (**E**) due to Kpn infection at the genetic level and promoted the anti-inflammatory factor IL-10 (**F**).

**Figure 8 ijms-24-15940-f008:**
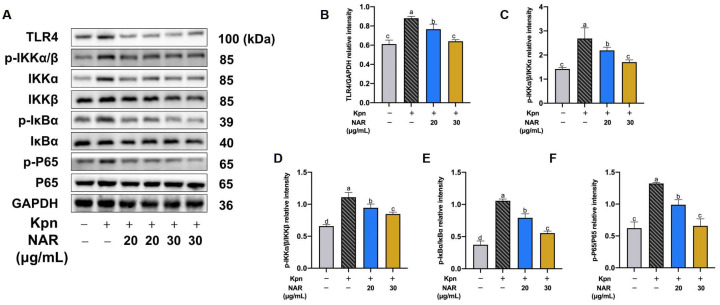
Effect of NAR on the NF-κB signaling pathway in MH-S. (**A**) Western blotting analysis of the NF-κB signaling pathway. Relative quantitative protein expression levels indicated that NAR intervention reduces the protein expression of p-IκBα/IκBα, p-IKKα-β/IKKα, p-IKKα-β/IKKβ, and p-P65/P65 in MH-S. (**B**–**F**) The protein expression of TLR4, p-IκBα/IκBα, p-IKKα-β/IKKα, p-IKKα-β/IKKβ, and p-P65/P65 in MH-S. One-way ANOVA statistical tests and Tukey’s post hoc analysis were used.

**Table 1 ijms-24-15940-t001:** Effect of NAR on blood biochemistry in mice.

	Control	NAR (80 mg/kg)	NAR (160 mg/kg)
ALB (g/L)	36.28 ± 1.34	37.42 ± 1.69	36.26 ± 1.20
ALP (U/L)	265.64 ± 19.42	256.18 ± 23.21	271.50 ± 30.64
ALT (U/L)	49.94 ± 7.36	46.19 ± 11.12	55.83 ± 14.06
AST (U/L)	158.97 ± 17.72	156.77 ± 26.52	165.93 ± 10.89
GREA (μmol/L)	34.72 ± 6.06	35.59 ± 8.14	38.32 ± 5.20
TBiL (μmol/L)	0.97 ± 0.13	0.83 ± 0.24	0.91 ± 0.14
TC (mmol/L)	2.87 ± 0.26	2.86 ± 0.33	2.97 ± 0.36
TG (mmol/L)	1.11 ± 0.19	1.26 ± 0.43	1.13 ± 0.27
TP (g/L)	77.34 ± 2.17	76.28 ± 4.27	76.10 ± 2.43
UREA (mmol/L)	9.13 ± 0.81	8.90 ± 0.82	9.41 ± 0.81

## Data Availability

The data presented in this study are available upon request from the corresponding authors.

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
