# Peer review of "Therapeutic Effect and Safety Evaluation of Naringin on Klebsiella pneumoniae in Mice"

_ijms, 2023, doi:10.3390/ijms242115940_

Round 1
Reviewer 1 Report
Comments and Suggestions for Authors
Review of the article "Therapeutic effect and safety evaluation of naringin on Klebsiella pneumonia in mice:" by Zhao et al.
As far as I know, there should be "Klebsiella pneumoniae" not "Klebsiella pneumonia" from the title throughout the article. May it be only dictionary's fault.
Well-written introduction. The description of the research is correct, enabling the research to be repeated by other research centers. The authors clearly present the purposefulness of the research undertaken and its background. Very well-planned and conducted research. I have no objections in this field.
However, I have a question why the authors did not use a currently available substance against Kpn to obtain information on whether NAR is actually better or worse than currently used drugs? The undoubted advantage of NAR is its natural origin and demonstrated therapeutic ability in the tested ranges.
A well-conducted discussion with numerous references to scientific articles on the research topic.
Line 377 - digits indicating the number of atoms should be in subscript.
One last question: why didn't the authors conduct research with the Kpn strain itself in laboratory conditions to find out how NAR works on Kpn?
Comments on the Quality of English LanguageMinor editing of English language required
Author Response
Dear Reviewer,
Thank you very much for spending a lot of time and effort to review our manuscript entitled "Therapeutic effect and safety evaluation of naringin on Klebsiella pneumoniae in mice" (Manuscript Number: ijms-2677381). We are also very grateful to for your valuable comments in order to improve the quality of our paper. We have marked the revised contents in red. The responses are as follows:
Question 1: As far as I know, there should be "Klebsiella pneumoniae" not "Klebsiella pneumonia" from the title throughout the article. May it be only dictionary's fault.
Response 1: Thank you for your careful inspection. “Klebsiella pneumoniae” is the correct spelling. We have corrected the misspelling "Klebsiella pneumonia" in the manuscript.
manuscript were as follows:
Page 1, line 3. Changed "Klebsiella pneumonia" to "Klebsiella pneumoniae"
Page 1, line 21. Changed "Klebsiella pneumonia" to "Klebsiella pneumoniae"
Question 2: However, I have a question why the authors did not use a currently available substance against Kpn to obtain information on whether NAR is actually better or worse than currently used drugs? The undoubted advantage of NAR is its natural origin and demonstrated therapeutic ability in the tested ranges.
Response 2: Thank you for your question. In clinical practice, broad-spectrum antibiotics such as carbapenems are mainly used to treat Kpn pneumonia. Obviously, the bacteriostatic effect of antibiotics is well recognized. However, we chose NAR as a research drug to develop NAR as a potential healthcare drug or food with enhanced anti-inflammatory ability. Therefore, in this study, we mainly demonstrated the anti-inflammatory effects of NAR at safe doses and did not compare the effects of NAR with those of commonly used antibiotics.
Question 3: digits indicating the number of atoms should be in subscript.
Response 3: Thank you very much for your suggestion. It has been revised in the manuscript.
manuscript were as follows:
Page 11 line 381 Changed "C27H32O14," to "C27H32O14"
Question 4: One last question: why didn't the authors conduct research with the Kpn strain itself in laboratory conditions to find out how NAR works on Kpn?
Response 4 Thank you very much for your suggestion. According to the literature, Ozcelik B and other teams[1-2] have investigated NAR's in vitro antibacterial activity against Kpn. In the present study research, the main focus was on the repairing effect of NAR on Kpn-induced inflammatory injury and related mechanisms, so the in vitro bacteriostatic assay was out of the scope of this study.
- Ozçelik B, Kartal M, Orhan I. Cytotoxicity, antiviral and antimicrobial activities of alkaloids, flavonoids, and phenolic acids. Pharm Biol. 2011 Apr;49(4):396-402. doi: 10.3109/13880209.2010.519390. Epub 2011 Mar 11. PMID: 21391841.
- Wang Z, Ding Z, Li Z, Ding Y, Jiang F, Liu J. Antioxidant and antibacterial study of 10 flavonoids revealed rutin as a potential antibiofilm agent in Klebsiella pneumoniae strains isolated from hospitalized patients. Microb Pathog. 2021 Oct;159:105121. doi: 10.1016/j.micpath.2021.105121. Epub 2021 Jul 31. PMID: 34343655.
Reviewer 2 Report
Comments and Suggestions for Authors
Dear authors, you presented an exciting paper considering the complications of coronavirus infection in the form of pneumonia. The paper considers the protective effect of naringinin (NAR) intervention against Kpn-induced pneumonia and the study of the safety of NAR . It is worth attention that the number of Kpn colonies in bronchoalveolar lavage fluid dropped significantly to 3.86, 2.07, and 0.27×107 CFU in response to NAR in the low-, medium- and high-dose groups, respectively. It indicated that Nar can be considered a medicinal product for preventing or even treating different complications of COVID-19.
The visualization of the paper is excellent (figures and pictures).
There are some remarks concerning improving the manuscript
1. In Lines 49-51 it is not clear what the authors mean
2. Line 134 - MPO - what is this&
3. Line 160-162. Very briefly I would add the sentence about phosphorylization, and its meaning in the inflammatory process
4. Line 372 it is not clear
5. There are no conclusions
6. 397 line - all the proc.....
7. 413 line - better to write oral administration
8. I would recommend to add a picture of Pummelo
9. Line 399 - very briefly it necessary to add that model group is infected without NAR (something like this)

it is very good
Author Response
Dear Reviewer
Thank you very much for spending a lot of time and effort to review our manuscript entitled "Therapeutic effect and safety evaluation of naringin on Klebsiella pneumoniae in mice" (Manuscript Number: ijms-2677381). We are also very grateful to for your valuable comments in order to improve the quality of our paper. We have marked the revised contents in red. The responses are as follows:
Question 1: In Lines 49-51, it is not clear what the authors mean
Response 1: Thank you very much for your suggestion. The sentence has been rewritten in the manuscript.
Manuscript were as follows:
Page 2, lines 47-56. Changed "Pummelo Peel (Hua Ju Hong) is mainly derived from Exocarpium Citri Grandis, the dried unripe fruit peel of Citrus grandis' Tomentosa' or Citrus grandis (L.). As a traditional Chinese medicine with the exact origin in medicine and food, it is mainly used to dissolve phlegm, remove dampness to benefit the stomach, and eliminate food, leading to increased use in daily health care. However, due to its low production and limited production areas, the general application of Pummelo Peel has been limited to a certain extent. However, flavonoids are the practical parts of Pummelo Peel, among which the content of dihydroflavonoid naringin (NAR) can reach up to more than 98%, which is the main active ingredient in Pummelo Peel. " to " Pummelo Peel (Hua Ju Hong) is mainly derived from Exocarpium Citri Grandis, the dried unripe fruit peel of Citrus grandis' Tomentosa' or Citrus grandis (L.), Which has the functions of dissolving phlegm, removing dampness to benefit the stomach, and eliminate food. As a medicinal and food homologous traditional chinese medicine, Pummelo Peel has been increasingly used in medicine and healthcare food. However, due to its low production and limited production areas, the general application of Pummelo Peel has been limited to a certain extent. However, flavonoids are the practical parts of Pummelo Peel, among which the content of dihydroflavonoid naringin (NAR) can reach up to more than 98% , which is the main active ingredient in Pummelo Peel "
Question 2: Line 134 - MPO - what is this&
Response 2: Thank you for your questions. MPO is an abbreviation for myeloperoxidase, and the full name has been added to the text already
manuscript was as follows:
Page 4, line 138 Changed "MPO" to "Myeloperoxidase(MPO)"
Question 3: Line 160-162. Very briefly I would add the sentence about phosphorylation, and its meaning in the inflammatory process
Response 3: Thank you for your suggestion. The meaning of phosphorylation for NF-κB has been added to the manuscript.
Manuscript were as follows:
Page 4, lines 168-170 Add the sentence: “The progressive phosphorylation of IKK, IκBα and P65 is a key step in the activation of the NF-κB signaling pathway, which can reflect the activation level of NF-κB.”
Question 4. Line 372 it is not clear
Response 4 Thank you for your questions. It has been revised and marked in the manuscript.
manuscript were as follows:
Page 14, lines 503-504 Changed "In conclusion, NAR is mild, can alleviate bacterial infections in the lungs, and can be a healthcare product against bacterial infections." to " In conclusion, NAR is a safe and effective drug to alleviate bacterial infections in the lungs and can be a plant-based health product that alleviates pneumonia."
Question 5.There are no conclusions
Response 5. Thank you for your suggestion. Conclusions have been supplemented in the manuscript.
manuscript were as follows:
Page 11 lines 496-504. In conclusion, the present study demonstrated that NAR enhanced the resistance to Kpn and attenuated the symptoms of Kpn-induced pneumonia in mice. Specifically, NAR reduced the phosphorylation of essential proteins of the macrophage NF-κB signaling pathway, decreased the aggregation of NEs in the lungs, lowered the expression of IL-6 and TNF-α, and attenuated lung tissue damage induced by the excessive inflammatory cytokines. Moreover, using NAR in effective doses is safe for the organism. In conclusion, NAR is a safe and effective drug to alleviate bacterial infections in the lungs and can be a plant-based health product that alleviates pneumonia. Add this paragraph from the discussion to the conclusion.
Question 6: 401 line - all the proc
Response 6: Thank you for your questions.
manuscript were as follows:
Page 12, line 401 "procedures" has been changed to "experimental process."
Question 7:413 line - better to write oral administration
Response 7 It has been modified to oral administration
manuscript were as follows
Page 12, line 420 Changed "administration" to "oral administration"
Question 8: I would recommend to add a picture of Pummelo
Response 8: Thank you for your suggestion. Picture with Pummelo have been included in the conclusions.
manuscript were as follows
Page 14, line 505
Page 15, lines 506-509
Add
Figure 9. The underlying mechanism of NAR attenuates bacterial infections induced by Kpn. NAR can inhibit the activation of the NF-κB signaling pathway in macrophages, reduce the recruitment of neutrophils, down-regulate inflammatory markers such as IL-6, TNF-α, and MPO, and alleviate lung injury caused by Kpn.
Question 9: Line 399 - very briefly it necessary to add that model group is infected without NAR (something like this)
Response 9: Thank you for your suggestion. A more complete explanation has been provided in the manuscript.
manuscript were as follows
Page 12, lines 403-408
Changed "(20 mg/kg), NAR medium-dose group (40 mg/kg), and NAR high-dose group (80 mg/kg) randomly (n=12)." to "A more complete explanation has been provided in the manuscript. Control group(Gavage with water), model group(Gavage with water + 108 CFU/mL Kpn infection), NAR low-dose group (Gavage with 20 mg/kg NAR + 108 CFU/mL Kpn infection), NAR medium-dose group (Gavage with 40 mg/kg + 108 CFU/mL Kpn infection), and NAR high-dose group (Gavage with 80 mg/kg + 108 CFU/mL Kpn infection) randomly (n=12)."